# EBBA-detector: An effective detector for defect detection in solar panel EL images with unbalanced data

Yixing Zhang[1,2]*, Ziyan Mo[3], Zhuan Xin[1], Xianyu Chen[1], Yuqin Deng[1], Xuan Dong[4]

**1** Geely University of China, Chengdu, Sichuan, China, **2** Sichuan Zhengruosheng Technology Company Limited, Chengdu, Sichuan, China, **3** Chengdu Xindu District Nanfeng primary school, Chengdu, Sichuan, China, **4** Sichuan Shenhong Chemical Industry Company Limited, Chengdu, Sichuan, China

* yixingzhangyz@163.com

## Abstract

Solar panel defect detection, a crucial quality control task in the manufacturing process, often faces challenges such as varying defect sizes, severe image background interference, and imbalanced data sample distribution. To address these issues, this paper proposes the EBBA-Detector. The core of the model lies in an enhanced balanced attention framework, which includes an Enhanced Bidirectional Feature Pyramid Network (EBFPN) and a Balanced-Attention Module (B-A Module). The EBFPN captures defect features of different sizes, significantly improving the recognition ability for small defects, while the B-A Module suppresses background interference, guiding the model to focus more on defect locations. Additionally, this paper designs a Scaled Dynamic Focal Loss (SDFL) function, which enables the model to pay more attention to minority and hard-to-identify defect samples under imbalanced data distribution. Through experimental validation on a large-scale electroluminescence (EL) dataset, the proposed method has achieved significant improvements in detection performance, with a mean Average Precision (mAP) of 89.85%, outperforming other models in multiple defect category detections. Therefore, the EBBA-Detector not only effectively detects small target objects but also demonstrates good handling capabilities for large targets and imbalanced data, providing an efficient and accurate solution for solar panel defect detection.

## 1. Introduction

Polycrystalline silicon solar cells have gained significant popularity due to their abundant raw materials, low cost, high conversion efficiency, and excellent stability, making them dominant in the solar panel market. However, the complex manufacturing process of polycrystalline silicon solar cells often leads to various defects during production. These internal and external defects not only reduce energy conversion efficiency but, if not detected and addressed in time, may severely damage photovoltaic

 

**Data availability statement:** The PVEL-AD dataset used in this study is available from the original publication: Su B, Zhou Z, Chen H. PVEL-AD: A Large-Scale Open-World Dataset for Photovoltaic Cell Anomaly Detection. IEEE Trans Ind Inform. 2023;19(1):404-13. doi: 10.1109/TII.2022.3162846. Researchers can access the dataset by following the data availability guidelines provided in the aforementioned publication.

**Funding:** The author(s) received no specific funding for this work.

**Competing interests:** The authors have declared that no competing interests exist.

modules, thereby threatening the safety and reliability of end-user products [1–3]. Therefore, defect detection in solar panels is essential. It helps manufacturers identify and eliminate defective panels in a timely manner, preventing them from entering the next production stage, thereby improving overall production efficiency. Additionally, it reduces the production of substandard products, lowering after-sales service and warranty costs. To promptly identify and mark panels that do not meet quality standards, improve production efficiency, and prevent defective products from entering the market, it is crucial to explore and develop more accurate methods for solar panel defect detection.

In real factory environments, solar panel inspection relies on a specialized defect detection system, which consists of four components: the supply subsystem, image acquisition subsystem, image processing subsystem, and sorting subsystem [4]. In the image acquisition subsystem, electroluminescence (EL) imaging is used to capture images. EL imaging is a non-destructive method commonly used for quality inspection of solar panels. By providing high-resolution EL images, the system achieves excellent defect detection capabilities. Solar panels are typically divided into two types: polycrystalline and monocrystalline silicon. Fig 1(a) and Fig 1(b) show a comparison of EL images of monocrystalline and polycrystalline silicon solar panels, highlighting crack and finger interruption defects.

In comparison, monocrystalline silicon solar panels have a more ordered crystalline structure, resulting in uniform luminescence distribution in their EL images, making defects easier to detect. In polycrystalline silicon solar panels, metal lines and the grid structure on the back surface often form significant dark areas in EL images, while dislocations and four busbars also appear as dark regions. The grain structure is more irregular, with grain boundaries and edges typically appearing as dark or mottled areas in the images. These factors collectively contribute to the complex background texture in EL images.

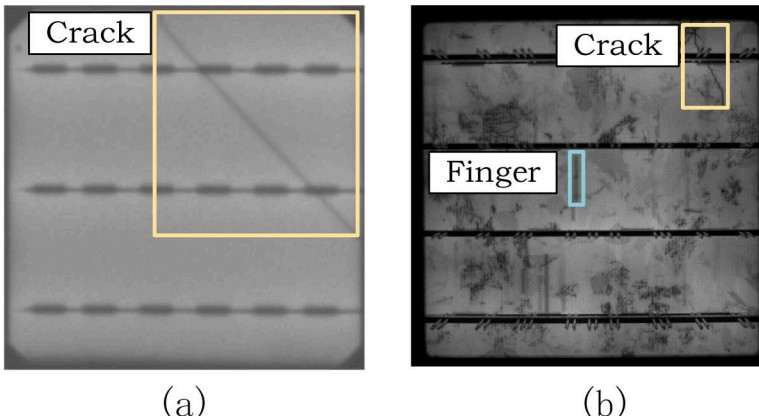

**Fig 1. EL image of photovoltaic cell.** (a) Monocrystalline photovoltaic cell (b) Multicrystalline photovoltaic cell.

Many methods have been proposed for solar panel defect detection. Traditional object detection methods typically include data classifier-based approaches and handcrafted feature extraction methods [5,6]. Data classifier-based methods use feature descriptors to generate feature vectors based on texture, color, shape, and spectral cues, followed by classifiers such as Support Vector Machines (SVM) to achieve defect detection. Handcrafted feature extraction methods rely on manually designed descriptors, often created by experts based on the characteristics of different defects, resulting in limited generalization and robustness.

In recent years, Convolutional Neural Networks (CNNs) have achieved remarkable performance in object detection tasks across various domains. Object detection networks include single-stage detectors such as SSD [7], the YOLO series [8], and two-stage detectors like Faster R-CNN [9]. Researchers have successfully applied these models to various defect detection tasks, such as leather defect detection [10], micro-LED defect detection [11], and PCB defect detection [12], achieving excellent results. These advancements have also opened new avenues for solar panel defect detection [13]. Although many object detection models have achieved some success in solar panel defect detection tasks, directly applying existing models to detect certain types of defects in solar panels may not be optimal. Analyzing EL images and defect characteristics reveals several specific challenges in solar panel defect detection:

**(1) Defects exhibit weak, small, and multi-scale characteristics**

Statistics indicate that the pixel proportion of finger interruption defects in the entire image ranges from 0.029% to 0.223%, while that of crack defects ranges from 0.011% to 0.409% (Fig 2(a) and Fig 2(b)). Clearly, both finger and crack defects fall into the category of small target defects and are embedded within complex backgrounds. In the field of solar cell defect detection, techniques for detecting large defects have become relatively mature and stable in performance. However, due to the downsampling effect during the continuous training process of neural networks, many small defect features are prone to being lost. Therefore, accurate identification and localization of small target defects remain challenging [14].

The statistical results of pixel area for finger interruption and crack defects, based on 100 randomly selected high-resolution original images with different defects, show that finger interruption defects account for 0.029% to 0.223% of the entire image, while crack defects account for 0.011% to 0.409%.

**(2) Images have complex non-uniform background textures**

In EL images, dislocations and four busbars appear as significant dark areas, and the grain structure presents irregular shapes randomly distributed in the image. Grain boundaries and edges usually appear as dark or mottled areas, leading

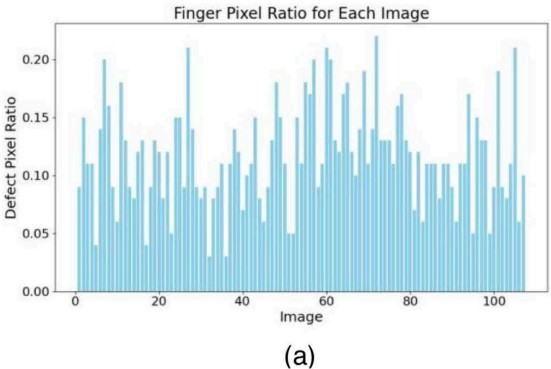
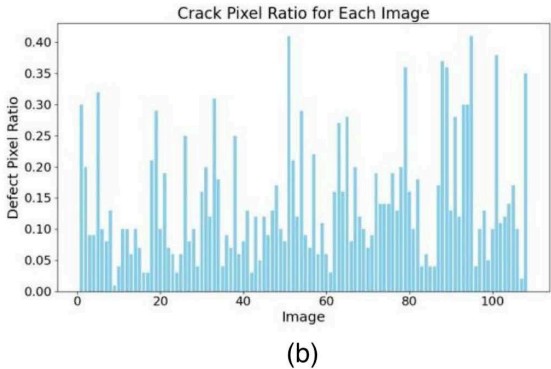

(a) (b)

**Fig 2. Statistical diagram of the proportion of finger and crack pixels in the whole image.**

to significant brightness differences between the background and defects. These factors contribute to the complex non-uniform background texture in EL images, adding extra difficulty to defect detection [15].

**(3)  Imbalanced class distribution in the dataset**

In the solar panel production process, the occurrence probability of different defects varies significantly. Specifically, finger interruption defects have a higher occurrence frequency, resulting in a relatively larger number of samples in the dataset. This class imbalance poses challenges for model training and performance evaluation. During training, the model may be biased towards majority classes due to their larger sample size, limiting the recognition ability for minority classes [16].

To address the first challenge, i.e., the loss of small target features due to network downsampling during training, Lin et al. [17] proposed the Feature Pyramid Network (FPN). FPN integrates feature information from different layers of the backbone network through upsampling and element-wise addition, enabling multi-scale feature fusion and enhancing the network's ability to recognize small targets. To preserve detailed shallow features, Liu et al. [18] further proposed the Path Aggregation Network (PANet), which improves the information flow in region-based instance segmentation frameworks. PANet enhances the bottom-up path, shortens the information flow distance between low-level and high-level features, and optimizes the overall feature hierarchy through precise localization signals, further improving the detection accuracy of small target defects.

To address the second challenge, i.e., background texture complexity, attention mechanisms are often used to suppress redundant background information. Building upon Faster R-CNN with FPN, Su et al. [4] proposed an improved BAFPN structure integrated with a multi-head cosine non-local attention module. This module enhances refined information of crack and broken grid defects across different scales, facilitating effective feature transmission and background interference suppression. However, BAFPN fails to sufficiently leverage attention modules to amplify defect features.

Regarding the data sample imbalance problem, Wang et al. [19] developed a photovoltaic cell defect detection method combining data augmentation and class weight allocation, which effectively mitigates performance degradation caused by insufficient data and class imbalance. A coordinate attention mechanism was incorporated into feature maps. As network depth increases during training, defect information of cracks and broken grids may be compressed during information propagation, potentially leading to feature loss. Chen et al. [20] addressed this issue by adopting an improved approach based on Focal Loss for multi-scale segmentation tasks.

To effectively suppress background noise while reinforcing and highlighting defect features, ensuring preservation of critical small-target characteristics during network deepening, the proposed EBBA (Enhanced Fused-Feature Bottleneck Attention) framework is introduced—a multi-scale network applicable for pyramid feature fusion. EBBA comprises two components: EBFPN and the B-A module, which respectively achieve enhanced defect feature prominence and background feature suppression. Furthermore, a novel loss function termed SDFL is proposed to address dataset imbalance, thereby improving detection performance for solar cell images. The main contributions of this work are summarized as follows:

(1) **Enhanced fusion module with dynamic balancing:** An enhanced fusion module is proposed, employing a bottom-up feature fusion strategy with lateral cross-layer connections. This fusion method can more effectively integrate shallow detail features with deep semantic features, helping to correctly represent multi-scale features. Considering the importance differences of different feature layers in detection tasks, learnable weights are introduced to dynamically balance information from each layer. Combined with the cosine non-local attention mechanism, the model's ability to filter background noise while detecting defects is enhanced. Experiments show that this method significantly improves the model's localization and object recognition accuracy.

(2) **Scaled Dynamic Focal Loss (SDFL):** To address inter-class sample imbalance in datasets and enhance model focus on hard-classified samples, a Scaled Dynamic Focal Loss (SDFL) is designed. Building upon Focal Loss, this

loss function introduces an adaptive weight adjustment mechanism that constrains weight values within a reasonable range to prevent over-amplification or suppression. The adaptation better suits photovoltaic cell defect detection tasks and improves detection accuracy.

(3) **EBBA-Detector for industrial applications:** The EBBA-Detector is developed, an object detector that embeds EFFA-FPN into the Region Proposal Network (RPN) of Faster R-CNN + FPN. Experimental results show that EBBA-Detector performs well in multi-scale defect detection in EL images, significantly improving the detection of multi-scale defects in EL images, meeting the high-quality requirements of photovoltaic cell industrial production.

The remainder of this paper is organized as follows. Section 2 provides an overview of related work. Section 3 details the proposed method. Section 4 presents extensive experiments and ablation studies. Section 5 discusses the experimental results. Finally, Section 6 concludes the paper.

## 2. Related methods

To extract multi-size defect features from the complex interference background of solar cells, experts and scholars have conducted a series of studies on solar cell defect detection, proposing some machine vision detection methods and deep learning detection methods.

Machine vision detection methods typically include filter-based methods [21] and handcrafted feature extraction methods. Filter-based methods mainly use image spectral information to extract texture features. Tsai et al. [22] proposed a Fourier image reconstruction method to identify defects such as cracks and finger interruptions in EL images. This method can effectively detect linear and strip-shaped defects. Handcrafted feature extraction methods require manual construction of complex recognition relationships. Su et al. [23] introduced a new feature descriptor, CPICS-LBP, by thresholding each pixel of the image into binary codes to fuse CPICS-LBP. Based on similarity analysis and clustering of image sample features, they proposed the CPICS-LBP bag, which achieved good classification results on the EL-2019 dataset. Du et al. [24] used Principal Component Analysis (PCA) to detect image data, establishing an eddy current thermography (ECT) system for Si-PV cells, enabling efficient and innovative defect detection on existing industrial production lines. This method has been successfully applied to silicon photovoltaic cell defect classification and detection tasks. Demant et al. [25] proposed a pattern recognition method based on local descriptors and support vector classification to detect micro-crack defects in photoluminescence (PL) and EL images. Machine vision-based defect detection has largely achieved automation, but the types of defects that can be detected are limited, and the accuracy is generally low. For different types and sizes of defects, carefully designed feature extraction methods are needed to effectively extract key features, requiring expertise and experience, thus lacking generality and adaptability.

With the increasing application of deep learning in industry, using deep learning for solar panel defect detection has gradually become a research direction for many scholars [26]. Single-stage object detection models like the YOLO series have the advantages of fewer parameters and fast detection speed. Two-stage object detection networks first generate candidate regions, then further classify and precisely locate these regions, usually achieving higher accuracy. In 2023, Li et al. [27] proposed a solar panel defect detection method based on YOLOv5 with a small target prediction head (GBH-YOLOv5), but the detection types are limited. Fu and Cheng [28] introduced an Efficient Long-Range Convolutional Network (ELCN) module by integrating ELCN into the YOLOv7 object detector for detecting defects in raw EL images, improving accuracy and reducing model parameters. In 2024, Cao et al. [29] used data augmentation to expand the dataset, improved the YOLOv8 backbone network, and replaced convolutions with GSConv, improving inference speed and reducing model size. Attention mechanisms help models focus on important information and suppress irrelevant information. Lu et al. [30] optimized YOLOv5 and introduced the CA attention mechanism to improve the model's feature extraction capability. Xie et al. [31] considered cross-domain detection, formulating it as an unsupervised domain adaptation problem, and used the SE attention module in the network. Zhang et al. [32] considered the visual features of defects

and noise in solar cells, using a global pairwise similarity module and a connected saliency module to refine features extracted by convolutional neural networks, proposing a global pairwise similarity and connected saliency-guided neural network. Adel Mellit [13] optimized deep conventional neural networks (DCNNs) and developed an embedded photovoltaic module fault detection and diagnosis system based on infrared thermography. F.L. et al. [33] proposed a semi-supervised anomaly detection model based on adversarial generative networks for solar panel defect detection. Su et al. [34] designed a Complementary Attention Network (CAN) for electroluminescence images, adaptively suppressing background noise features and highlighting defect features while leveraging the complementary advantages of channel and spatial location features. They embedded CAN into the region proposal network of Faster R-CNN to extract more precise defect region proposals, detecting cracks, finger interruptions, and black cores. Chen et al. [35] proposed a Bidirectional Path Feature Pyramid Network (BPFPN) and a Group Attention Module (GAM). Liu et al. [36] proposed a Boundary Refinement Module (BRM) that employs a learnable module $I_{offset}$ to predict two offset values for each boundary point based on high-frequency features ($HF$), effectively improving detection accuracy. Liu et al. [37] adopted geometric transformations such as translation and rotation on raw data to augment minority-class datasets, alleviating data imbalance issues. Experimental results demonstrate that this approach achieves relatively favorable detection performance. However, the artificially augmented data generated through such transformations inherently deviates from fully authentic raw data, which may limit the model's generalization capability in real-world complex scenarios. Consequently, there remains a need to design loss functions with enhanced generalizability and adaptability to ensure robust performance across diverse data distributions. Su et al. [4] improved the BAFPN structure based on Faster RCNN+FPN, using a multi-head cosine non-local attention module to emphasize crack and finger interruption defect features. However, this method did not achieve ideal results for crack defect detection and did not address data imbalance, motivating this research.

## 3. Methodology

In this section, we first introduce the overall framework of the EBBA-Detector. We then delve into the EBBA Network, which consists of two main components: the EBFPN module and the B-A module. Finally, we provide a detailed description of the proposed loss function, SDFL.

### A. EBBA-detector model architecture

To effectively utilize attention mechanisms while considering model performance and computational resources, we propose the EBBA-Detector for solar panel defect detection. The model comprises three parts: (A) a feature extraction component that extracts meaningful feature representations from input images; (B) a region proposal component responsible for generating candidate target regions or bounding boxes; and (C) a component that classifies and regresses each candidate region. The overall architecture of the EBBA-Detector is based on Faster R-CNN, with the EBBA-RPN integrated into the RPN. Fig 3 illustrates the overall process of the solar panel defect detection model.

Specifically, in the first part, the input solar panel image is resized from 1024 × 1024 pixels to 600 × 600 pixels before being fed into the defect detection model. To better adapt to the dataset characteristics, ResNet-101 is trained from scratch as the backbone for feature extraction. The outputs from multiple lower layers of the ResNet-101 network are selected to construct a pyramid-shaped feature hierarchy (C2, C3, C4, and C5), which retains both high semantic information and lower resolution information. Since C1 occupies a large amount of memory, it is excluded. The second part involves further feature extraction using the proposed EBBA-RPN. Finally, in the third part, the relationship between the original image and the feature map is used to generate matrices, which are then fed into the RoI pooling layer. RoI pooling maps each RoI to the corresponding position in the feature map, divides each RoI region into several equal-sized parts, and performs max-pooling to generate a fixed-size feature map. These features are then processed by the classification prediction layer and the bounding box regression layer: the classification prediction layer predicts the possible object categories in each RoI, while the bounding box regression layer predicts the coordinate offsets of the bounding

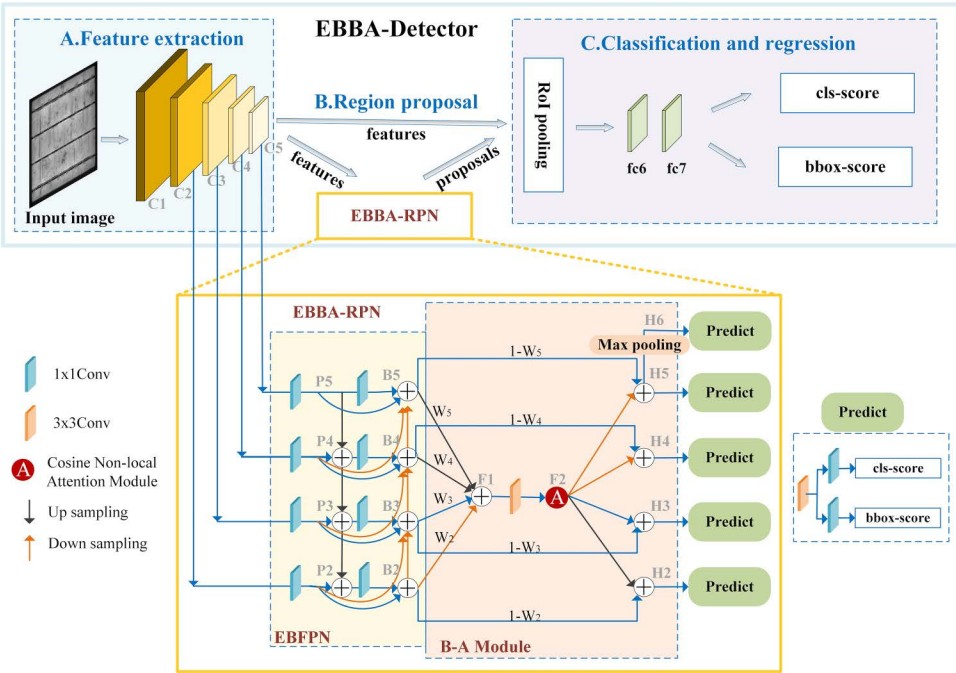

**Fig 3. EBBA-Detector model structure diagram.**

boxes for each RoI. The total loss function, which includes the proposed Scaled Dynamic Focal Loss (SDFL) and the bounding box regression loss, is used for backpropagation to update the model parameters via stochastic gradient descent (SGD). After obtaining the classification and bounding box regression outputs, non-maximum suppression (NMS) is applied to eliminate overlapping detection results, and a threshold is set to filter out low-confidence detections. The final output consists of two key parts: (1) a C × 1 confidence vector, where C represents the number of categories in the target dataset, and each element represents the score for the corresponding category, with the most likely category determined by the Argmax function; and (2) the bounding box information ($x_{min}$, $y_{min}$, $x_{max}$, $y_{max}$) and score information for each category.

Having introduced the overall architecture of the EBBA-Detector and the training process for the solar panel defect detection model, we now delve into the key factors for model performance optimization and accuracy improvement: the EBBA-RPN module and the design of the loss function.

## B. Enhanced bidirectional balanced-attention network

Various fusion schemes have emerged, but the combination of a bottom-up path with cross-layer fusion significantly enhances the features of small target defects and better adapts to targets of different scales and sizes, thereby improving the model's generalization ability. In existing research, Su et al. [4] directly applied attention mechanisms to refine the outputs of only the middle two layers of the network. Typically, applying attention mechanisms to as many layers as possible allows the model to focus on different positions when processing input data, thereby capturing global information more comprehensively. However, this approach often comes with high computational costs. To effectively utilize attention mechanisms, we propose the B-A module. Unlike previous methods that independently add attention modules to each layer, this module learns to fuse features from different layers, reducing computational costs while demonstrating superior background suppression effects.

## 1) Enhance Bidirectional FPN

As shown in Fig 4(a), the EBFPN module aggregates multi-scale features at different resolutions. The module first generates feature maps P2, P3, P4, and P5 from the feature hierarchy C2, C3, C4, and C5 using FPN. The top-down and bottom-up feature fusion processes are described by the following formulas:

$$p'_i = \begin{cases} conv_{1\times1}(C_i), & if\ i = i_{max} \\ upsample(P'_{i+1}) + conv_{1\times1}(C_i), & if\ i < i_{max} \end{cases} \tag{1}$$

$$P_i = conv_{3\times3}(P'_i) \tag{2}$$

where $C_i \in \mathbb{R}^{H\times W\times C}$ (C, W, H represent the number of channels, width, and height of the feature map, respectively) denotes the feature map output from the i-th layer of the ResNet-101 network, $P_i' \in \mathbb{R}^{H\times W\times C}$ represents the feature map generated by the top-down path after feature fusion, and $P_i \in \mathbb{R}^{H\times W\times C}$ represents the corresponding feature map generated by applying a 3×3 convolution to $C_i$. The upsampling operation, as shown in Fig 5(a), uses bilinear interpolation to enlarge the feature map from the previous layer, which is then added to the side feature map after a 1×1 convolution. To reduce aliasing effects caused by upsampling, a 3×3 convolution is applied, completing the top-down path. Bilinear interpolation estimates the value of a new pixel using the weights of the surrounding four pixels. The bottom-up path is then added to generate feature maps B2, B3, B4, and B5. The fusion method for the bottom-up path is described by the following formula:

$$B_i \begin{cases} conv_{1\times1}(P_i) + downsample(B_{i-1}) + downsample(P_{i-1}), & if\ i = i_{max} \\ conv_{1\times1}(P_i) + conv_{1\times1}(C_i), & if\ i = i_{min} \\ conv_{1\times1}(P_i) + downsample(B_{i-1}) + downsample(P_{i-1}) + conv_{1\times1}(C_i), & if\ i_{min} < i < i_{max} \end{cases} \tag{3}$$

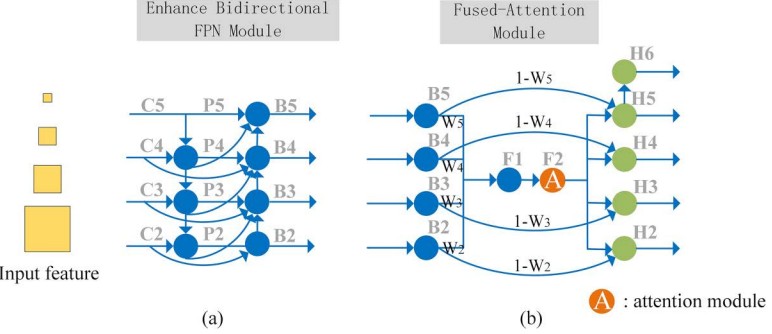

**Fig 4. EBBA-RPN Framework (a) EBFPN (b) B-A Module.**

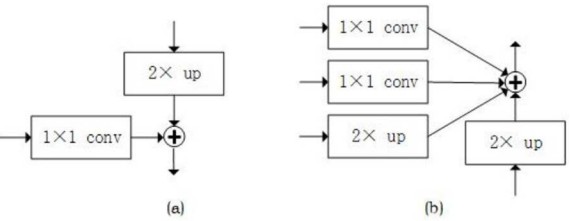

**Fig 5. (a) Upsampling (b) Downsampling.**

where $B_i \in R^{H \times W \times C}$ represents the feature map generated by the bottom-up path after multi-level feature fusion. The downsampling process, as shown in Fig 5(b), uses max-pooling. The EBFPN module aggregates multi-scale features at different resolutions.

## 2) Balanced-Attention Network

Since different layers contribute differently to object detection, learnable weights w2, w3, w4, and w5 are introduced to balance the information from each feature layer, allowing the model to automatically learn the importance of each feature layer and better capture the original feature information, rather than simply averaging the features from multiple layers. The Softmax function is then applied to the learned parameters, normalizing the values between 0 and 1. This strategy aims to adjust the impact of each layer on the final detection results based on its relative importance. The feature maps B2, B4, and B5 are resized to match the size of B3 ($76 \times 76 \times 256$) through upsampling and downsampling, and the resized feature maps are multiplied by $w2$, $w3$, $w4$, and $w5$ and summed to obtain the refined feature map $F1 \in R^{H \times W \times C}$.

$$F1 = w_5 \times \text{upsample}(B_5) + w_4 \times \text{upsample}(B_4) + w_3 \times B_3 + w_2 \times \text{downsample}(B_2) \tag{4}$$

After applying a $3 \times 3$ convolution to F1, the cosine non-local attention mechanism is introduced, as shown in Fig 6. First, the input feature map $F_1 \in R^{H \times W \times C}$ is fed into the cosine non-local attention module and transformed into an $HW \times C$ matrix, where each pixel's feature vector is stacked by channel to form a row vector. For each position in the transformed feature map, the attention score is obtained by calculating the similarity between that position and other positions, resulting in a similarity matrix $HW \times WH$. This study uses cosine similarity to compute the attention score. For each position $i$ in the input feature map $x$, the cosine similarity $f(i,j)$ with position $j$ is calculated as follows:

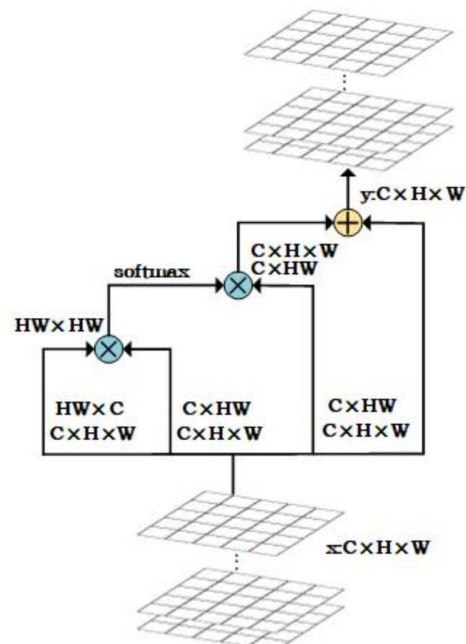

**Fig 6. Architecture of the proposed cosine nonlocal attention module.**

$$f(i,j) = \frac{x_i^T x_j}{\|x_i\| \, \|x_j\|} \tag{5}$$

where $x_i$ and $x_j$ represent the feature vectors at positions $i$ and $j$, respectively. The numerator is the dot product of the vectors, and the denominator represents the magnitudes of the vectors.

After obtaining $f(i,j)$, the Softmax operation is applied for normalization, as shown in Equation 6. In this formula, Softmax$(i,j)$ represents the output value after applying the Softmax function to $f(i,j)$, indicating the importance of each pixel in the feature map. The denominator sums the exponential values of the cosine similarities across all positions $(k,l)$ in the feature map, serving as a normalization factor to ensure that the sum of Softmax$(i,j)$ is 1, resulting in a probability distribution. If dot product similarity were used to compute $f(i,j)$, the denominator in the Softmax formula could become very large, causing the final result to approach zero, indicating minimal differences between elements. Therefore, cosine similarity is used to compute the attention score, ensuring that $f(i,j)$ ranges between $[-1, 1]$.

$$Softmax(i, j) = \frac{exp(f(i,j))}{\sum_{k=1}^{H} \sum_{l=1}^{W} \exp\left(f\left(i, (k, l)\right)\right)} \tag{6}$$

The Softmax operation assigns weights to each pixel based on its cosine similarity with other pixels. These weights are multiplied by the transposed feature map to obtain a $C \times HW$ matrix, representing the attention-weighted feature representation for each channel. Finally, the weighted feature representation is reshaped to $C \times H \times W$ and added to the input feature map to obtain the final output feature map $F_2 \in R^{C \times H \times W}$. This process is described by the following formula:

$$F2 = \text{Att}\left(\text{conv}_{3 \times 3}(F1)\right) \tag{7}$$

where Att() represents the cosine non-local attention mechanism. Higher weights are assigned to important feature layers, guiding the model to focus more on the layers critical for solar panel defect detection. The enhanced features are then adjusted to the original multi-scale features, forming the final multi-level features. To more comprehensively capture feature information, the multi-scale feature maps B2, B3, B4, and B5 are multiplied by $1 - wi$ and fused with the corresponding output features to obtain H2, H3, H4, and H5. This operation indicates that for important layers, more attention is given to the feature map after applying the attention mechanism, while for less important layers, more of the original feature information is retained. Finally, the feature map H5 undergoes max-pooling to generate an additional feature map H6. This process is described by the following formula:

$$
\begin{aligned}
H_i &= (1 - w_i) \times B_i + F_i, \quad if \ i = 2, \ 3 \\
H_i &= (1 - w_i) \times B_i + downsample\,(F_i), \quad if \ i = 4, \ 5 \\
H_6 &= Maxpooling\,(H_5)
\end{aligned} \tag{8}
$$

## C. Scaled dynamic focal loss

In the EBBA-Detector, two loss functions are used for training: classification loss and regression loss. In the second stage, many solar panel defect detection models directly use Cross-Entropy (CE) loss for classification loss. For easy-to-classify samples, $p_i$ is close to 1, and when the sample is correctly classified, the loss approaches 0. For hard-to-classify samples, $p_i$ may be small, and log($p_i$) approaches negative infinity when the sample is misclassified. This causes the loss term for hard-to-classify samples to be biased towards smaller values, resulting in insufficient learning for these samples. Focal Loss was proposed to address the foreground-background class imbalance in single-stage detectors. In multi-class scenarios, Focal Loss can also be used to handle class imbalance. We propose the Scaled Dynamic Focal Loss (SDFL), which improves the Focal Loss by adding a weight factor that not only adjusts the contribution of class weights to the loss but also considers the stability of the training process through overall scaling. The formula is as follows:

$$SDFL(p, u) = -\sum_{u=1}^{k} \alpha_u (1 - p_u)^{\gamma} y_u log(p_u)$$

$$\alpha_u = m \left(1 - \frac{count(x_u)}{count(x)}\right)^{\gamma} \tag{9}$$

where $count(x_u)$ is the number of samples with label $u$ in the dataset, $count(x)$ is the total number of samples in the dataset, and $m$ is the maximum value of the weight scaling range. For example, if the total number of solar panel defect samples in the training set is 7747, and the number of samples labeled as "star crack" is 101, then $\alpha_u = 3.896$. Through multiple experiments, we found that setting $m = 4$ and $\gamma = 2$ achieved the best model performance. By adjusting the weights with a common $\gamma$, the model's handling of different classes and hard-to-classify samples becomes more consistent, reducing sensitivity to different factors during training. The overall second-stage loss formula is as follows:

$$L(p, u, t^u, v) = L_{cls}(p, u) + \lambda[u \geq 1]L_{loc}(t^u, v) \tag{10}$$

$$L_{loc}(t^u, v) = \sum_{i \in \{x,y,w,h\}} p_u^* smoothL_1(t_i^u, v_i)$$

$$L_{cls}(p, u) = SDFL(p, u) \tag{11}$$

In Formulas (10) and (11), u represents the true class label, $u \in 0, 1,..., K$. p represents the predicted probability that the i-th anchor box is the true label. In multi-class scenarios, p represents the Softmax probability distribution predicted by the classifier $p = (p_0, p_1, ..., p_k)$. When $u \geq 1$, $[u \geq 1]$ evaluates to 1; otherwise, it evaluates to 0. This part indicates that regression loss is only used when there is a defect, and background categories do not require regression loss. $t^u$ represents the regression parameters for the bounding box of the i-th anchor box predicted from the original image $t_i = [t_x, t_y, t_w, t_h]$. v represents the true bounding box regression parameters for the i-th defect $t_i^* = [t_x^*, t_y^*, t_w^*, t_h^*]$. $\lambda$ is the balance parameter, set to 1, to balance the two losses. The formula for smooth$_{L1}$ is as follows:

$$smoothL_1 \begin{cases} 0.5(t_i - t_i^*)^2, & if |t_i - t_i^*| < 1 \\ |t_i - t_i^*| - 0.5. & otherwise \end{cases} \tag{12}$$

Thus, SDFL not only considers overall scaling for training stability but also accounts for the influence of class frequency and sample prediction probability, assigning higher weights to samples with lower frequency but higher prediction probability $p_u$. Fig 7 provides pseudo-code describing the training process of the proposed method.

## 4. Experiments

This section details the dataset used, key implementation details, and conducts extensive ablation studies to comprehensively evaluate the performance of the proposed EBBA-Detector. Subsequently, the EBBA-Detector is compared with state-of-the-art defect detection methods, and its superior performance and achievements are discussed. Additionally, Grad-CAM visualization is used to intuitively demonstrate the improvements of the EBBA-Detector, supporting the experimental results and helping to better understand the superior performance of the EBBA-Detector, showcasing its unique advantages and application value in the field of defect detection.

---

**Algorithm 1 :** The Detailed Training Procedure of the Model

**Input:** The training sample set $T = x_1, x_2, ..., x_n$, where
$i = 1, 2, ..., n$, and each $x \in R^d$.

**Output:** Obtain the best model parameters $W_{best}$.

1: % Signal meanings:
2: % $W$- the weight matrix
3: % $c$ - an index, a group of parameters in the weight matrix
4: % $W_c^{(t)}$ - $W$ at the $t$-th iteration
5: % $W_c^{(t+1)}$ - $W$ at the $t + 1$-th iteration
6: % $\eta$ - the learning rate
7: % $\nabla L_{train}(W_c^{(t)})$ - the gradient of the loss function $L_{train}$ with respect to the weight matrix $W_c^{(t)}$
8:
9: **Begin**
10: % Randomly initialize the model parameters $W \in R^d$.
11: **while** not done **do**
12: %Train the network parameters using the target dataset.
13: %The weights are updated using a stochastic gradient Descent (SGD) solver.
14: % Update the weight parameters
15: $W_c^{(t+1)} = W_c^{(t)} - \eta \cdot \nabla L_{train}(W_c^{(t)})$
16: %The Scaled Dynamic Focal Loss (SDFL) function is utilized. SDFL refers to equation (3-12).
17: % Continuously update the weight parameters $W$ using the sample set $T$
18: **end while**
19: **End**

---

**Fig 7. The Detailed Training Procedure of the Model.**

## A. Dataset

This paper conducts a series of experiments on the publicly available PVEL-AD dataset [38] to evaluate the performance of the proposed method.

In the field of multi-crystalline solar panel EL image defect detection, this dataset is used for benchmarking solar panel anomaly detection methods. The investigation focuses on five defect categories, with cracks, finger interruptions, and black core defects representing the most prevalent types. Notably, cracks and finger interruptions exhibit extremely low pixel proportions of 0.011% and 0.029% relative to the entire image, categorizing them as small-sized targets. In contrast, star cracks and thick line defects occur less frequently, reflecting the class-imbalanced distribution inherent in the dataset.

The images in the PVEL-AD dataset have a resolution of 1024 × 1024. To ensure the comprehensiveness and accuracy of the experiments, we used a 6:2:2 split ratio for training, validation, and testing sets (details provided in Table 1).

The PVEL-AD dataset (Fig 8) provides ground-truth bounding box annotations for defects, with a focus on defect diversity and detection challenges. To comprehensively evaluate model performance, we curated representative defect samples including:

(1) Structurally distinct defects (e.g., black cores and star cracks);

**Table 1. Dataset partitioning by defect category.**

| Category | Training Set | Validation Set | Test Set |
|---|---|---|---|
| Finger | 1812 | 486 | 660 |
| Crack | 1042 | 288 | 256 |
| Blackcore | 296 | 98 | 112 |
| Star crack | 87 | 26 | 32 |
| Thick line | 183 | 61 | 85 |

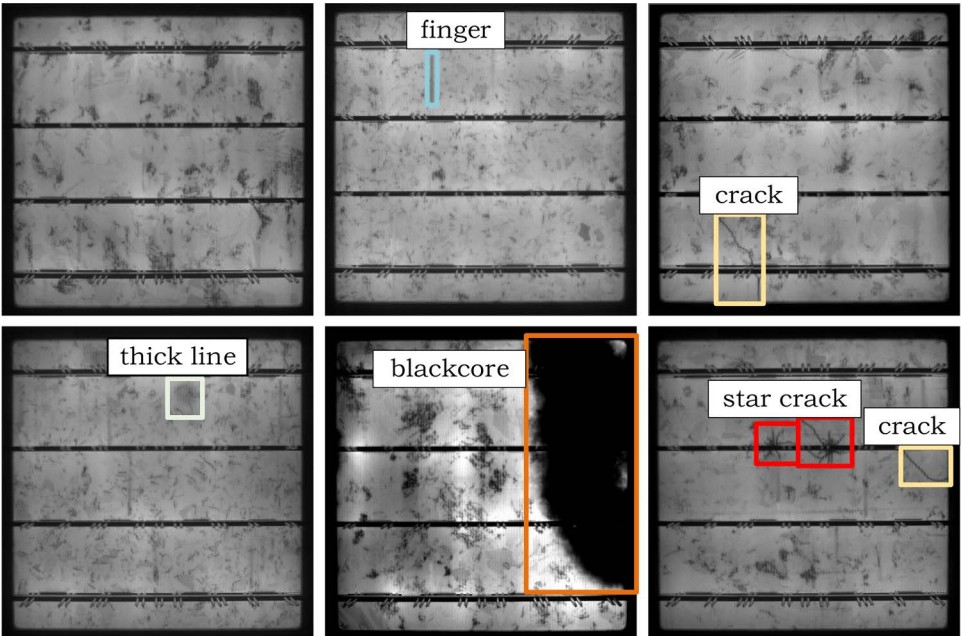

**Fig 8. PVEL-AD samples.**

(2) Texturally ambiguous defects exhibiting high similarity to background patterns (e.g., cracks, finger interruptions, and thick lines);

(3) Composite defect samples containing co-occurring defect categories to rigorously assess localization and classification accuracy under multi-target interference.

This systematic selection enables precise performance evaluation of the novel defect detection methodology across diverse defect morphologies and challenging scenarios.

## B. Evaluation Metrics

To evaluate the performance of different methods in solar panel EL image defect detection, we employ Average Precision (AP) and mean Average Precision (mAP) as performance metrics. The formulas for precision (P) and recall (R) are as follows (Equations 13 and 14):

$$P = \frac{TP}{TP + FP}$$

(13)

$$R = \frac{TP}{TP + FN} \tag{14}$$

where TP (True Positive) represents the number of true defects correctly predicted as defects; TN (True Negative) represents the number of true non-defects correctly predicted as non-defects; FP (False Positive) represents the number of true non-defects incorrectly predicted as defects; and FN (False Negative) represents the number of true defects incorrectly predicted as non-defects.

AP is used to measure the detection accuracy for each defect class, calculated as the area under the Precision-Recall curve. The Intersection over Union (IoU) threshold is set to 0.5 to determine valid detections. Here, **C** represents the number of defect categories. The mAP is the average of AP values across all solar panel defect categories, and it is used to evaluate the overall performance of different detection models. The formulas for AP and mAP are as follows (Equations 15 and 16):

$$AP = \sum_{i=1}^{n} P_i \left( R_i - R_{i-1} \right) \tag{15}$$

$$mAP = \frac{1}{C} \sum_{i=1}^{C} AP_i \tag{16}$$

These metrics provide a comprehensive assessment of the model's ability to accurately detect and classify defects in solar panel EL images.

## C. Implementation details

This study employs Pytorch and MMdetection for experimentation. To ensure the effectiveness of the proposed EBBA-Detector, all training, testing tasks, and comparative experiments described in this paper were conducted using a GPU with 16 GB of memory (NVIDIA RTX 3050). The solar panel image dataset consists of images with a resolution of $1024 \times 1024$ pixels, which are resized to $600 \times 600$ pixels and expanded to three channels before being input into the detection model. During model training, the initial learning rate is set to 0.001, and the maximum number of iterations is 150. A step decay strategy is adopted, where the learning rate is reduced at specific intervals. Specifically, the learning rate is multiplied by 0.1 every 15,000 and 30,000 iterations. Due to GPU memory limitations, the batch size is set to 1. A prediction box is considered a positive sample if its Intersection over Union (IoU) with any ground truth box exceeds 0.7. Non-Maximum Suppression (NMS) is applied to remove duplicate prediction boxes, with the maximum number of detections set to 1000. The EBBA-Detector's parameter count is 62.74 million, with an inference speed of 6.2 FPS. The detailed hyperparameters for training and testing the EBBA-Detector are listed in Table 2.

**Table 2. Hyperparameters for training and testing.**

|  | Hyperparameter | Value |
|---|---|---|
| Training | Weight_decay | 0.0001 |
|  | Batch_size | 1 |
|  | Lr | 0.001 |
|  | Decay_step | 15000, 30000 |
|  | Momentum | 0.9 |
|  | Base_anchor_size_list | [32,64,128,256,512] |
| Testing | Nms_iou_threshold | 0.5 |
|  | Show_score_threshold | 0.3 |

The changes in loss values during the training of the proposed EBBA-Detector are shown in <u>Fig 9(a)</u> and <u>Fig 9(b)</u>, which records both classification loss and regression loss.

In the first 1000 iterations, the network converges rapidly, with the loss value decreasing sharply. This is likely due to the sensitivity of parameter adjustments in the initial stages and the ease with which the model captures the features of the training data. After 2500 iterations, the rate of loss reduction slows, indicating that the model is approaching the optimal solution, with less room for parameter adjustments, leading to slower improvements in the loss value. After 10,000 iterations, the loss curve stabilizes, indicating that the model has converged to a relatively stable state, and further training allows the model to better adapt to variations and noise in the training data. Overall, the model exhibits good convergence, with the stability of the loss values meeting expectations.

## D. Ablation studies

To evaluate the effectiveness of each component of the proposed network, we conduct ablation experiments on the PVEL-AD dataset, the results of which are shown in <u>Table 3</u>.

The ablation study results demonstrate that the proposed EBFPN module and B-A module collectively improve the model's mean Average Precision (mAP) by 3.8% compared to the baseline architecture, thus validating the effectiveness of the EBBA Framework in small target detection. Furthermore, the integration of SDFL (Sparse-Dense Focal Loss) achieves an additional 1.4% mAP gain in solar panel defect detection, highlighting its capability to enhance recognition performance for rare defect patterns.

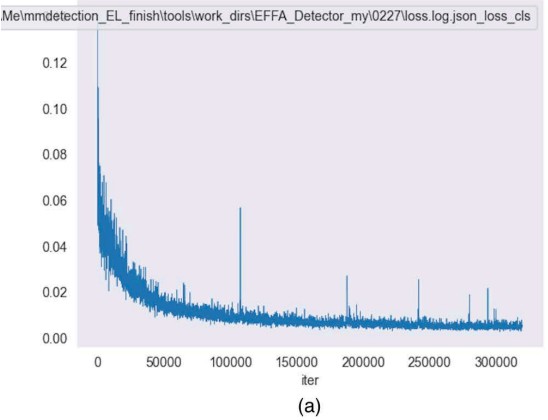
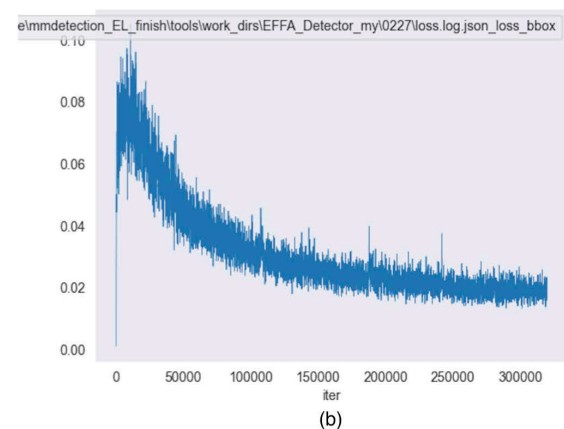

(a)  (b)

**Fig 9. Change graph of loss value (a) Classification loss (b) Regression loss.**

**Table 3. Ablation experiment results.**

| Faster R-CNN | EBFPN | B-A Module | SDFL Loss Function | Crack AP(%) | Finger AP(%) | mAP @0.5(%) |
|---|---|---|---|---|---|---|
| √ | | | | 78.3 | 91.0 | 84.65 |
| √ | √ | | | 80.1 | 91.2 | 85.65 (+1.0) |
| √ | | √ | | 80.6 | 93.1 | 86.85 (+2.2) |
| √ | √ | √ | | 83.8 | 93.1 | 88.45 (+3.8) |
| √ | √ | √ | √ | 84.9 | 94.8 | 89.85 (+5.2) |

### 1) Effect of EBBA framework

The proposed framework is specifically designed to enhance small-target detection accuracy under complex backgrounds. A meticulously engineered cross-layer fusion mechanism within the feature pyramid architecture is introduced to strengthen the model's capability in representing solar panel defect features. To further highlight key features and suppress irrelevant information, we introduce the B-A Module, enabling the model to adaptively focus on the most critical features for defect detection. To verify the effectiveness of the EBFPN and B-A Module, we conducted experiments embedding each module separately.

First, when only the EBFPN is embedded, the model significantly improves the mean Average Precision (mAP) to 85.65% by fully utilizing multi-scale feature information. This result demonstrates the effectiveness of EBFPN in enhancing feature representation. Second, when only the B-A Module is used, the mAP improves by 2.2%. This improvement indicates that the model can more effectively focus on key features through the attention mechanism, thereby improving defect detection accuracy. When both EBFPN and B-A Module are applied, the mAP improves by 3.8%. T Experimental results demonstrate that the synergistic integration of the EBFPN and B-A Module significantly enhances feature representation for small-target defects. The proposed EBBA framework effectively improves detection accuracy for miniature targets by enabling the model to comprehensively leverage multi-level feature hierarchies, achieving precise identification of solar cell defects under cluttered backgrounds.

### 2) Effect of SDFL

The proposed loss function is specifically designed to address rare defect detection challenges. Its adaptive weight adjustment mechanism dynamically recalibrates class-specific weights during training, effectively countering imbalanced data distributions inherent in solar panel defect detection tasks. Experimental results demonstrate a 5.2% improvement in

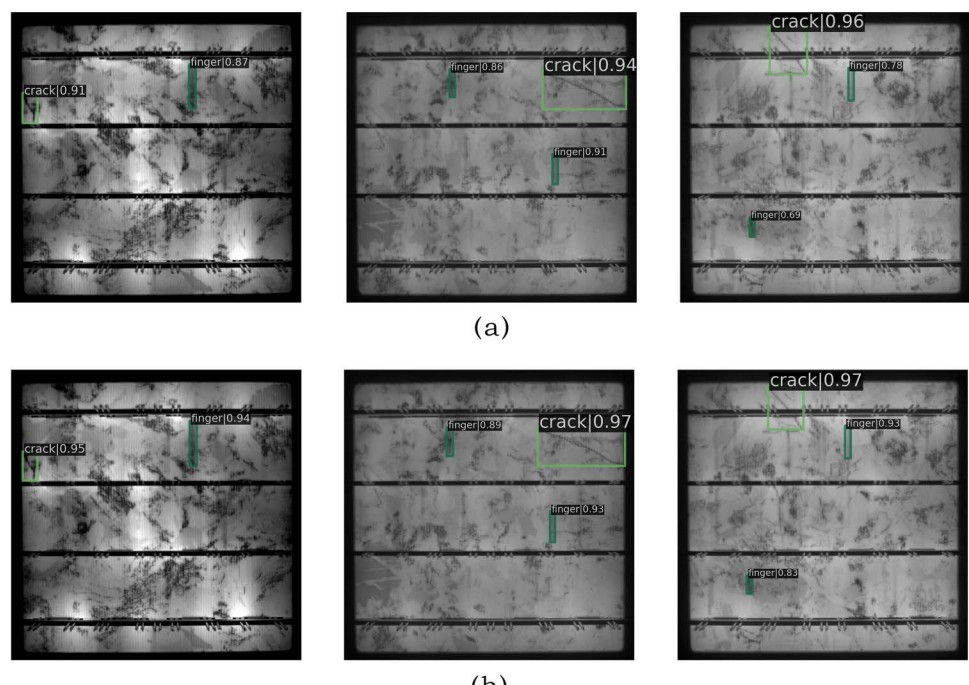

(a)

(b)

**Fig 10. Detection results (a) Before using SDFL (b) After using SDFL.**

mAP compared to the baseline model. As shown in Fig 10, the implementation of the Scaled Dynamic Focal Loss (SDFL) significantly enhances detection confidence for critical defects such as finger interruptions and micro-cracks. This loss function enables the model to comprehensively learn features from all defect categories, particularly those with limited samples. The designed loss exhibits strong generalizability, being applicable not only to the EBBA-Detector framework but also extensible to other defect detection scenarios with class imbalance.

### 3) Overall Evaluation

To verify the effectiveness of the proposed method, Fig 11 shows the Class Activation Map (CAM) for crack and finger interruption defects in solar panel EL images. The CAM is a visualization tool.

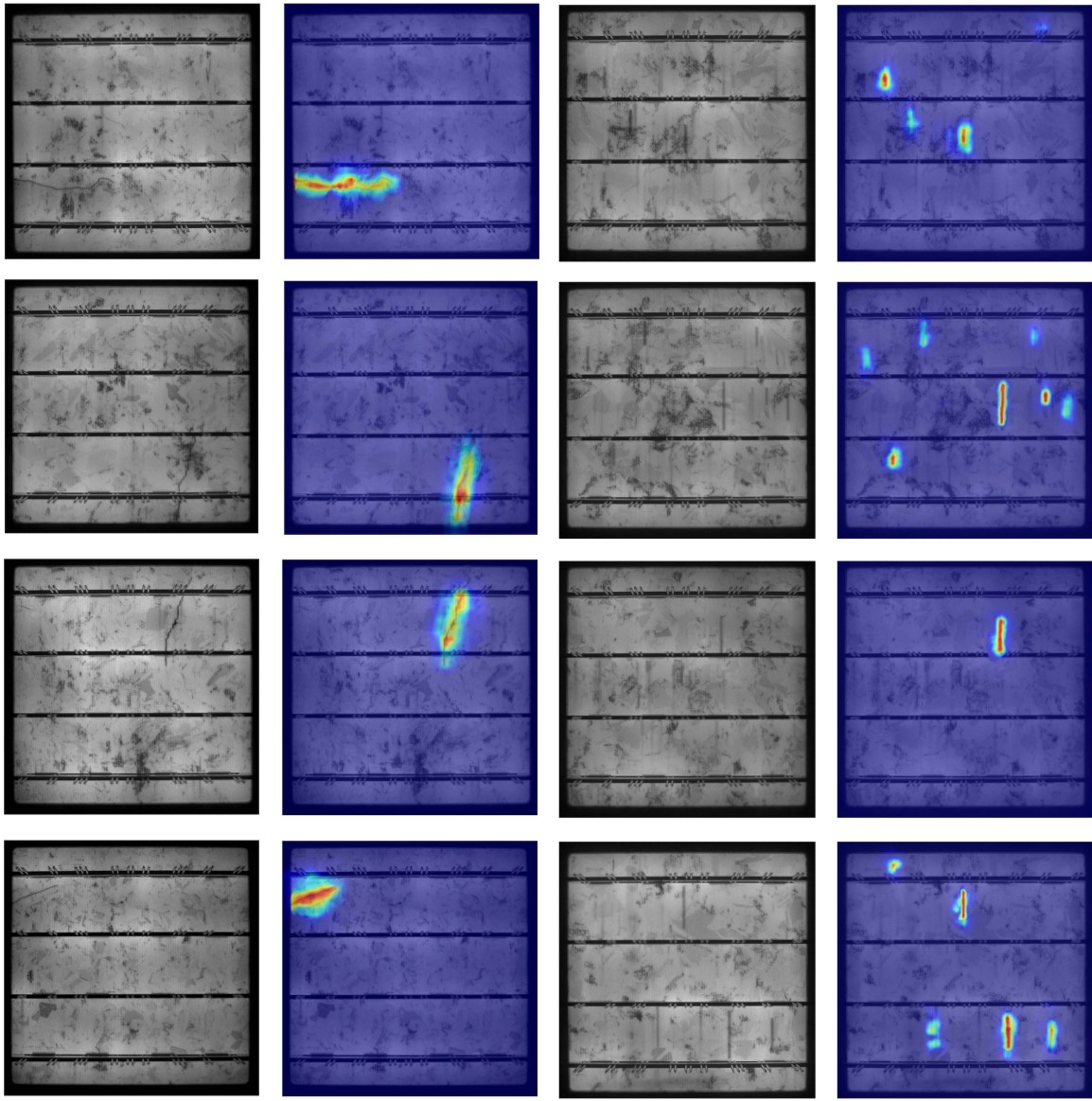

**Fig 11. Display of CAM.**

In the CAM, the red highlighted areas indicate the regions where the model focuses most on defects, corresponding to the locations of finger interruptions and cracks in the original image. Yellow and light blue areas indicate lower activation levels or blurred boundary regions, representing the model's lower attention to the target categories. The shorter and blurrier finger interruptions in the upper part of the original image appear as blue in the CAM. This demonstrates that the proposed model has excellent feature extraction capabilities and can effectively locate crack and finger interruption defects.

## 5. Discussions

To evaluate the superiority of the proposed method, we reproduced several mainstream deep convolutional neural network-based object detection methods and conducted comprehensive comparative experiments under the same dataset and experimental conditions. The comparison methods include single-stage object detection models (SSD [7], VFNet [39], Foveabox [40], RetinaNet [41], Mobilenetv2 [42], HRNet [43], YOLOv8 [44], YOLOv9 [45]) and two-stage object detection models (Cascade R-CNN [46], Faster R-CNN [9], BAF-Detector [4]). The experimental results are shown in Table 4.

The YOLO series represents classic single-stage object detection algorithms. In YOLOv8, released in 2023, YOLOv8-x has the largest number of parameters. Analysis of the detection results on this dataset shows that YOLOv8-x achieves a 3.35% higher mAP than YOLOv8-m. The release of YOLOv9 on February 22, 2024, marks further improvements in detection performance. From Table 4, it can be seen that YOLOv9 achieves the highest AP for finger interruption defects in this experiment, reaching 95.5%, but its recall rate is lower than that of YOLOv8. SSD and VFNet have higher recall rates, but their mAP is slightly lower compared to two-stage object detection methods. Therefore, for complex or small targets, two-stage object detection methods still have advantages. Although the EBBA-Detector's AP for finger interruption defects is slightly lower than that of YOLOv9, its AP for crack defects is significantly higher, surpassing YOLOv9 by 6.3% and Cascade R-CNN by 4.7%. In terms of mAP and recall rate, the EBBA-Detector outperforms the original BAF-Detector, with the highest mAP among all models, reaching 89.85%. This indicates that the EBBA-Detector can more accurately locate and identify targets, reducing false positives and improving the reliability of detection results.

By observing the improvements in accuracy for crack and finger interruption defects, we further conducted experiments on five common defect categories in solar panel EL images, including the large target category black core. The comparison of detection results for the five defect categories is shown in Table 5.

**Table 4. Comparison of different model detection results for two types of defects.**

| Model Category | Model Name | Crack AP(%) | Finger AP(%) | mAP @0.5(%) | Recall (%) |
|---|---|---|---|---|---|
| Single-stage | SSD | 79.5 | 90.5 | 84.9 | 96.8 |
| | YOLOv8-m | 73.6 | 94.4 | 84.0 | 81.5 |
| | YOLOv8-l | 77.4 | 94.8 | 86.1 | 81.1 |
| | YOLOv8-x | 79.3 | 95.4 | 87.35 | 80.8 |
| | YOLOv9-c | 78.6 | 95.5 | 87.1 | 76.3 |
| | YOLOov9-e | 73.0 | 95.3 | 84.2 | 78.4 |
| | VFNet | 79.8 | 93.8 | 86.8 | 96.6 |
| | Foveabox | 73.9 | 88.2 | 81.0 | 96.0 |
| | RetinaNet | 63.8 | 88.7 | 76.2 | 96.2 |
| | HR-Net | 79.1 | 90.3 | 84.7 | 95.9 |
| Two-stage | Casade R-CNN | 80.2 | 93.8 | 87.0 | 93.5 |
| | Faster R-CNN | 78.3 | 91.0 | 84.65 | 94.9 |
| | BAF-Detector | 79.8 | 91.0 | 85.4 | 95.9 |
| | EBBA-Detector | **84.9** | 94.8 | **89.85** | **97.0** |

**Table 5. Comparison of different model detection results for five types of defects.**

| Model name | crack AP(%) | finger AP(%) | black core AP(%) | Star crack AP(%) | thick line AP(%) | mAP@ 0.5(%) | Recall (%) |
|---|---|---|---|---|---|---|---|
| SSD | 73.2 | 90.4 | 98.9 | 61.2 | 78.7 | 80.5 | 92.2 |
| YOLOv8-m | 80.6 | 94.8 | 99.3 | 83.0 | 87.2 | 88.98 | 83.4 |
| YOLOv8-l | 80.9 | 94.6 | 97.9 | 85.5 | 86.6 | 89.1 | 82.6 |
| YOLOv8-x | 79.9 | 94.9 | 99.3 | 81.9 | 87.4 | 88.68 | 84.2 |
| YOLOv9-c | 71.8 | 94.6 | **99.4** | 89.4 | 88.6 | 88.76 | 81.0 |
| YOLOv9-e | 78.1 | 95.0 | **99.4** | 85.0 | **89.8** | 89.46 | 82.7 |
| VFNet | 78.5 | 93.3 | 97.7 | 79.2 | 87.6 | 87.3 | 96.4 |
| Foveabox | 81.5 | 90.1 | 98.9 | 83.1 | 87.3 | 88.2 | 96.5 |
| RetinaNet | 76.0 | 90.2 | 98.8 | 79.2 | 88.0 | 86.5 | 96.5 |
| HR-Net | 79.7 | 93.6 | 98.9 | 83.8 | 86.8 | 88.56 | 96.6 |
| CasadeR-CNN | 79.4 | 93.8 | 98.9 | 83.2 | 86.1 | 88.3 | 94.6 |
| Faster-rcnn | 76.7 | 92.0 | 98.7 | 79.9 | 84.7 | 86.4 | 96.0 |
| BAF-Detector | 78.3 | 91.8 | 99.0 | 79.7 | 84.3 | 86.6 | 96.0 |
| EBBA-Detector (CE) | 80.2 | 92.9 | 99.0 | 84.1 | 86.3 | 88.5 (+1.9) | 96.5 |
| EBBA-Detector (SDFL) | **86.1** | **95.1** | 98.9 | **90.7** | 87.7 | **91.7 (+5.3)** | **96.8** |

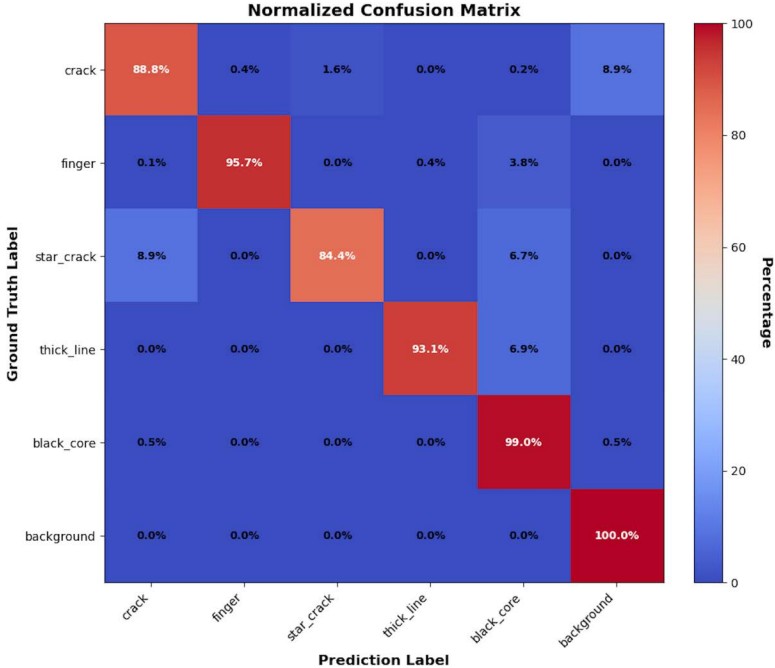

**Fig 12. Confusion matrix.**

From the experimental results, it can be seen that, except for black core, the AP for other categories improves to varying degrees. The improved model achieves a 5.3% increase in mAP compared to the original model, indicating that the improved model can more accurately identify targets in images. The recall rate improves by 0.8%, indicating that the improved model can better capture targets in images, reducing missed detections and improving the model's comprehensiveness and coverage. After applying the SDFL loss function, the mAP increases from 88.5% to 91.7%. The AP for black

core decreases by 0.1%, possibly because black core targets are larger and more distinct, making them easier to classify. For star crack, which has the fewest samples and less distinct features, the AP improves by 11%. Although YOLOv9-c achieves the highest AP for black core and thick line defects, it has the lowest accuracy for detecting small target defects such as crack. Therefore, the EBBA-Detector not only effectively detects small target objects but also demonstrates excellent performance in handling large targets and imbalanced data, significantly improving the accuracy of solar panel defect detection. Based on these results, we conclude that the EBBA-Detector exhibits significant advantages in solar panel defect detection tasks.

Fig 12 shows the confusion matrix used to evaluate the model's classification performance. Among all tested samples, none of the thick line defects are misclassified as other defects. Due to the high complexity of the background, a small portion of each defect is still classified as background. Specifically, 8.9% of star crack defects are misclassified as crack, and 1.6% of crack defects are misclassified as star crack. This is because the two defects have similar appearances, making it difficult for the model to distinguish between them.

By analyzing the confusion matrix results in conjunction with Table 5, it can be observed that the detection accuracy for crack defects is generally low, possibly due to variations in the size and shape of cracks, which depend on the depth, width, and morphology of the defects. In contrast, the detection accuracy for finger interruption defects is relatively high,

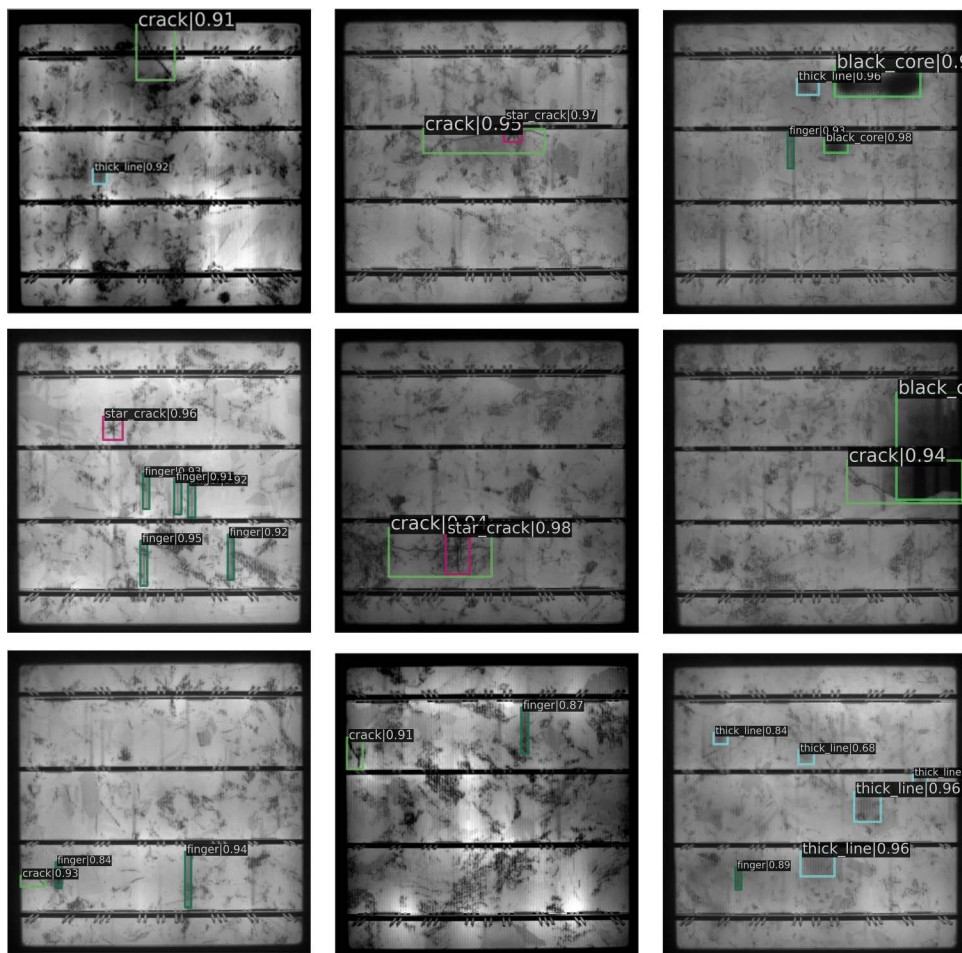

**Fig 13. EL image detection results of photovoltaic cell of different categories.**

as the morphology of these defects is simpler, making it easier for the detection algorithm to understand and identify them. Moreover, the training data contains more samples of finger interruption defects, allowing the model to better learn the features of these defects and thus improving detection accuracy. Compared to small target defects, black core defects typically have larger sizes and higher signal-to-noise ratios, making it easier for the algorithm to distinguish them from the background.

Fig 13 shows the detection results of the proposed EBBA-Detector for different categories of solar panel EL images. The EBBA-Detector demonstrates high detection accuracy and confidence across various categories, providing strong support for precise and reliable defect detection in solar panel EL images. Therefore, the proposed EBBA-Detector is highly effective for solar panel defect detection.

## 6. Conclusion

In this paper, the EBBA-Detector is proposed for solar panel EL image defect detection. The FBFPN and B-A Module are introduced, which complement the bottom-up feature transfer path, achieving deep fusion of cross-layer features and integrating attention mechanisms into a single balanced feature map. This combination significantly enhances the model's ability to represent small target defect features, enabling it to better understand and utilize multi-level feature information, thereby improving the model's discriminative performance in detecting defects of different scales and achieving optimization in both accuracy and speed. To address the issue of class imbalance in the dataset, the SDFL loss function is proposed, which greatly increases the model's attention to minority and hard-to-classify samples, enabling more precise defect localization and classification. Through extensive experiments and ablation studies on the PVEL-AD defect dataset, the effectiveness of the proposed method is validated, particularly in handling the common issues of class imbalance and complex background interference in solar panel EL image datasets. In extensive comparative evaluations with other object detection models, EBBA-Detector demonstrates significant advantages across various quantitative metrics. These results indicate that the proposed deep learning model has unique advantages in the field of defect detection, providing a practical solution for research and application in photovoltaic cell defect detection.

The widespread adoption of solar panel technologies has created an urgent need for advanced defect detection methods. While the proposed EBBA-Detector demonstrates promising performance in identifying photovoltaic defects, two critical limitations require attention. First, the model shows reduced effectiveness in detecting crack-type defects, primarily due to their diverse morphological characteristics and complex geometric variations. Second, the static parameterization of the loss function limits adaptability across different inspection scenarios. To address these challenges, future research will prioritize the development of geometry-aware neural architectures incorporating multiscale deformable attention mechanisms for enhanced crack characterization, alongside the implementation of meta-learned dynamic loss functions that autonomously optimize parameters ($m$, $\gamma$) based on real-time task complexity metrics. These improvements aim to significantly enhance detection accuracy while maintaining robust performance across varying industrial inspection environments.

## Acknowledgments

The authors would like to thank the anonymous reviewers for their valuable comments and suggestions, which helped improve the quality of this paper.

## Author contributions

**Conceptualization:** Yixing Zhang.

**Data curation:** Ziyan Mo.

**Formal analysis:** Yixing Zhang.

**Investigation:** Ziyan Mo, Xuan Dong.

**Methodology:** Yixing Zhang.

**Project administration:** Yixing Zhang, Zhuan Xin.

**Resources:** Ziyan Mo, Xuan Dong.

**Software:** Ziyan Mo.

**Validation:** Xianyu Chen.

**Visualization:** Yixing Zhang, Yuqin Deng.

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
