## [Decision Letter · Decision Letter 0]

PONE-D-25-11855EBBA-Detector: An effective detector for defect detection in solar panel EL images with unbalanced DataPLOS ONE

Dear Dr. Zhang,

Thank you for submitting your manuscript to PLOS ONE. After careful consideration, we feel that it has merit but does not fully meet PLOS ONE’s publication criteria as it currently stands. Therefore, we invite you to submit a revised version of the manuscript that addresses the points raised during the review process.

**ACADEMIC EDITOR's comments: ** The manuscript requires revision and rereview according to the reviewer comments. Additionally, few other comments are:

The images are not clear to check the results of detection. Provide high quality and clear figures. If the images quality is deteriorated during uploading as separate figures, then they can be included inside the manuscript file (set high quality option in text editor) for review purpose.

Add number of images/instances of each defect class in different data splits to show how rare and smaller defects and data imbalance is handled.

Focus more on detection results of rare and smaller defects. Add and discuss related results. How they are improved? As this is the main point addressed by this study.

PVEL-AD dataset is larger in size. How images are selected?

We look forward to receiving your revised manuscript.

Kind regards,

Muhammad Waqar Akram, PhD

Academic Editor

PLOS ONE

Additional Editor Comments:

The manuscript requires revision and rereview according to the reviewer comments. Additionally, few other comments are:

The images are not clear to check the results of detection. Provide high quality and clear figures. If the images quality is deteriorated during uploading as separate figures, then they can be included inside the manuscript file (set high quality option in text editor) for review purpose.

Add number of images/instances of each defect class in different data splits to show how rare and smaller defects and data imbalance is handled.

Focus more on detection results of rare and smaller defects. Add and discuss related results. How they are improved? As this is the main point addressed by this study.

PVEL-AD dataset is larger in size. How images are selected?

Reviewers' comments:

Reviewer's Responses to Questions

**Comments to the Author**

1. Is the manuscript technically sound, and do the data support the conclusions?

Reviewer #1: Yes

Reviewer #2: No

2. Has the statistical analysis been performed appropriately and rigorously? 

Reviewer #1: Yes

Reviewer #2: N/A

3. Have the authors made all data underlying the findings in their manuscript fully available?

Reviewer #1: Yes

Reviewer #2: Yes

4. Is the manuscript presented in an intelligible fashion and written in standard English?

Reviewer #1: Yes

Reviewer #2: No

5. Review Comments to the Author

Reviewer #1: This paper proposes the EBBA-Detector to address challenges such as varying defect sizes, severe image background interference, and imbalanced data sample distribution.

This work is valuable, but I still have some suggestions as follows:

(1) The authors need to clarify the contribution of this paper to the field and how it differs from other similar works in the introduction.

(2) In the literature review section, it is recommended to include more recent relevant studies. Suggested references are as follows:

Reference:

[1] A real-time anchor-free defect detector with global and local feature enhancement for surface defect detection[J]. Expert Systems with Applications, 2024, 246: 123199.

[2]An efficient CNN-based detector for photovoltaic module cells defect detection in electroluminescence images[J]. Solar Energy, 2024, 267: 112245.

(3) The authors should provide the number of parameters and the inference speed of the proposed method.

(4) The conclusion section can be further expanded to discuss the limitations of the proposed method and suggest more specific directions for future research.

Reviewer #2: Fig 1, 3, 8, 10, 11, 13: Replace with high-resolution images.

Tables 3–4: Adjust column widths, align decimals, and highlight critical results.

Confusion Matrix (Fig 12): Improve color contrast and label readability.

Training Curves (Fig 9): Ensure axis labels and legends are clear.

The manuscript’s technical content is strong, but it is not acceptable in its current format. Address the visual clarity issues and resubmit. If the figures are extracted from low-quality sources, regenerate them directly from the original data or software (e.g., MATLAB, Python plots). For tables, use LaTeX formatting or spreadsheet tools to ensure professionalism.

Data Presentation: Numerical results in tables and figures (e.g., AP/mAP values) should be bolded or highlighted for key comparisons. Confusion matrices (Fig 12) need sharper color contrast and labels.

Consistency: Ensure all figures follow the same style (font, axis scales, etc.) and are referenced properly in the text.

6. PLOS authors have the option to publish the peer review history of their article (what does this mean? ). If published, this will include your full peer review and any attached files.

**Do you want your identity to be public for this peer review?** For information about this choice, including consent withdrawal, please see our Privacy Policy .

Reviewer #1: **Yes: ** Min Liu

Reviewer #2: No

---

## [Author Response · Author response to Decision Letter 1]

7 May 2025

Dear Academic editor and Reviewers,

Thank you for your constructive feedback and the opportunity to revise our manuscript. We have carefully addressed all the comments. Below, we provide a point-by-point response to each concern.

Academic Editor’s Comments:

1. Image Clarity

All figures have been regenerated using high-resolution sources.

2. Data Splits for Rare/Small Defects

A new Table 1 in Section 4.A details the number of instances per defect class across training, validation, and test sets. In Section 4.D, based on the results of the ablation experiments, we have described in detail how our proposed method handles rare and smaller defects, as well as data imbalance issues. This provides a more comprehensive understanding of our approach to dealing with these challenges.

3. Discussing Improvements in Detecting Rare and Smaller Defects

We have expanded the conclusion discussion in Section 4.D. By analyzing each proposed module, we have highlighted the positive impacts on the detection results of rare and smaller defects. This analysis clearly demonstrates how our method improvements contribute to better detection performance for these types of defects.

4. PVEL-AD Dataset Selection

Section 4.A now includes a detailed description of the image selection criteria for the PVEL-AD dataset. This clarifies how we managed the large - scale dataset and selected appropriate images for our study.

Reviewer #1’s Comments:

1. Clarify Contributions in Introduction

We have updated the introduction section to clearly state the improvements over current research and highlight our main work. This revision makes it easier for readers to understand the unique contributions of our paper to the field, as well as the differences from other similar studies.

2. Include Recent Literature

Three more recent research including the suggested references ([19], [36] and [37]) have been added, with a discussion of their relevance to our work.

3. Model Parameters and Inference Speed

We have added the model's number of parameters and inference speed in Section 4.C. This information helps readers better evaluate the performance and efficiency of our proposed method.

4. Expand Conclusion with Limitations

In the "Conclusion" section (Section 6), we have further expanded the content to discuss the limitations of our proposed method. Additionally, we have provided more specific directions for future research, which will guide further exploration in this area.

Reviewer #2’s Comments:

1. Figure Quality (Figs. 1, 3, 8, 10, 11, 13)

For Fig 1, 3, 8, 10, 11, 13, we have resubmitted clear and high-resolution images.

2. Table Formatting (Tables 3–4)

We have adjusted the column widths of Tables, aligned the decimals, and highlighted the critical results. This makes the data in the tables more organized and easier to read.

3. Confusion Matrix (Fig. 12)

We have enhanced the color contrast and label readability of the confusion matrix (Fig 12).

4. Training Curves (Fig. 9)

We have ensured that the training curve (Fig 9) has clear axis labels and legends by resubmitting a high-quality version.

5. Data Presentation

Key numerical results are highlighted in yellow in Tables 4–5.

6. Consistency

All figures now follow a unified style and are referenced appropriately in the text.

We appreciate the reviewers’ thorough evaluation and believe these revisions significantly strengthen the manuscript. Please contact us if further clarifications are needed. We look forward to your positive feedback and the opportunity to have our work published in PLOS ONE.

Sincerely,

Yixing Zhang

Corresponding Author

Email: yixingzhangyz@163.com

---

## [Editor Report · Decision Letter 1]

EBBA-Detector: An effective detector for defect detection in solar panel EL images with unbalanced Data

PONE-D-25-11855R1

Dear Dr. Zhang,

We’re pleased to inform you that your manuscript has been judged scientifically suitable for publication and will be formally accepted for publication once it meets all outstanding technical requirements.

Kind regards,

Muhammad Waqar Akram, PhD

Academic Editor

PLOS ONE
---

## [Editor Report · Acceptance letter]

PONE-D-25-11855R1

PLOS ONE

Dear Dr. Zhang,

I'm pleased to inform you that your manuscript has been deemed suitable for publication in PLOS ONE. Congratulations! Your manuscript is now being handed over to our production team.

Kind regards,

on behalf of

Dr. Muhammad Waqar Akram

Academic Editor

PLOS ONE